# China's historical record in the search of tropical cyclones corresponding to ITCZ shifts over the past 2ka

Huei-Fen Chen[1,2], Yen-Chu Liu[1], Chih-Wen Chiang[1], Xingqi Liu[3], Yu-Min Chou[4], Hui-Juan Pan[1]

1. Institute of Earth Sciences, National Taiwan Ocean University, Keelung, Taiwan, R.O.C.
2. Center of Excellence for Oceans, National Taiwan Ocean University, Keelung, Taiwan, R.O.C.
3. College of Environmental Resources & Tourism, Capital Normal University, Beijing, P.R. China
4. Department of Ocean Science and Engineering, Southern University of Science and Technology, Shenzhen, P.R. China

## Abstract

The northwestern Pacific Ocean and south China sea are where tropical cyclones occur most frequently. Many climatologists also study the formation of Pacific Ocean warm pools and typhoons in this region. This study collected data of paleotyphoons found in China's official historical records over the past two thousand years with known typhoon activity reports. The collected data is then subjected to statistical analyses focusing on typhoon activity in coastal regions of southeastern China to garner a better understanding of the long-term evolution of moving paths and occurrence frequency, especially those typhoons making landfall in mainland China. We analyzed the data with the year and month of each typhoon event, as well as the number of events in a ten-year period. The result shows that (1) north/southward migration of typhoon paths correspond to the north/southward migration of the Intertropical Convergence Zone (ITCZ) during Medieval Warm Period (MWP) and Little Ice Age (LIA), (2) paleotyphoons made landfall in mainland China one month earlier during MWP than those during LIA. This implies a northward shift in ITCZ during MWP. Typhoons tend to make landfall in Japan during El Nino-like periods and strike the southern coastal regions of China during La Nina-like stages. According to paleotyphoon records over the last two thousand years, typhoons made landfall in southeastern China frequently around 490-510 A.D., 700-850 A.D., and after 1500 A.D. The number of typhoons striking Guangdong Province peaked during the coldest period in 1660-1680 A.D.; however, after 1700 A.D., landfall has migrated farther north. The track of tropical cyclones (TCs) in the northwestern Pacific Ocean is affected by the North Atlantic Oscillation (NAO) and the Pacific Decadal Oscillation (PDO), which shows a nearly 30-yr and a 60-yr cycle during the LIA.

Key word: Tropical cyclone, record, landfall, ITCZ, MWP, LIA, NAO

## 1. Introduction

Tropical cyclones (TCs) are a serious hazard. According to the Federal Emergency Management Agency (FEMA) of the USA, the total amount of money spent on flood recovery programs due to TC activity was greater than that spent on any other natural catastrophe during the period 2005 to 2015. The level of destruction caused by TCs has meant they have been the focus of a great deal of current research as well as part of the

historical record of China for millennia.

Among all tropical cyclones, 37% occur in the northwestern Pacific Ocean (Liang and Ye, 1993). These TCs are of a greater intensity and frequency of making landfall in this region than those making landfall in western Atlantic Ocean. People urgently give attention to the frequency and tracks of TCs on the earth. The path of TCs in Pacific Ocean is driven by the clockwise rotation of the North Subtropical Pacific High and it takes 3 paths away from this genesis region: (1) a westerly path straight toward south China; (2) a west-northwesterly path recurving to Japan; and (3) a north-oriented path that keeps them out to sea (Elsner and Liu, 2003). Most existing TC records are based on short-term researches that cover the past few decades (Wu and Lau, 1992; Lander, 1994). Short-term weather records indicate that TC paths may be directly influenced by variations of the El Niño Southern Oscillation (ENSO) in the equatorial Pacific region (Chan, 1985; Lander, 1994; Elsner and Liu, 2003; Ho *et al*., 2004; Chu, 2004), and ENSO is highly related to the PDO (Pavia et al., 2006; Feng and Wang, 2013). However, climate study literature is severely lacking longer-term studies with more data in hundreds of years. Another dynamic forcing influence the pathways of TCs is related to the ITCZ position and North Atlantic Oscillation (NAO) (Gil et al., 2006). For the purpose to track TC pathways in a long-term period, we need the geological records via the evidences of natural sediment from lake cores and lagoons in widespread coastal regions. The geological records indicate ancient TC activity were enhanced by the ENSO activity after middle Holocene, both in Atlantic and Pacific Oceans (Donnelly and Woodruff, 2007; Woodruff et al., 2009; Chen et al., 2012; McCloskey and Liu, 2012, 2013; McCloskey et al., 2013; Liu et al., 2015). In this study, we attempted to collate statistics on the landfall frequency of TCs recorded in China's written historical record with typhoon intensity recorded in the geological record of lake sediments in northeastern Taiwan to investigate TC path migration in the northwestern Pacific Ocean region over the last 2 ka.

## 2. Paleotyphoon records from China's official historical documents

A research discussed statistical records of regional TCs occurrence since 1851 from the southeastern coastal region of the United States of America in Atlantic Ocean regions (Bossak et al., 2014). Moreover, the historical record of TC occurrence in the northwestern Pacific owns longer historical records in China. Chan and Shi (2000) first published the frequency of typhoon landfall over Guangdong Province of China during the period of 1470 A.D.~ 1931 A.D, and then Liu et al. (2001) made an examination of historical records dating back to 1000 years ago in the Guangdong Province. A further research also tried to integrate statistical records of TC occurrence in southeastern costal China over the last 400 years (Fogarty, 2004). Therefore, we attempted to collect more TC data from these documents and understand some defect fragments in historical records.

China's historical record is a rich source of documented evidence on climatic conditions dating back millennia. Anomalous abnormalities in climatic conditions found in China's records had been successfully applied in the reconstruction of regional climate changes (Liu et al, 2001; Chu et al., 2002; Chu et al., 2008). Previous research revealed the term "Jufeng" (cyclone, 颶風) first appeared in the South-North Dynasty around 420-479 A.D. (Liu et al., 2001). During the following Tang Dynasty (618-907 A.D.) many climate phenomena relating to torrential rainfall and strong winds resembling typhoons were recorded in poems (Louie and Liu, 2003). After the Northern

Song Dynasty (960-1126 A.D.), Chinese governmental institutions have kept a continuous record of typhoon strikes reported by local administrative authorities (Louie and Liu 2003, Liu et al. 2003). The term "Typhoon" (颱風) first appeared during the Qing Dynasty with documented evidence of typhoon landfall on Taiwan first appearing in 1750 A.D.

### 3. Applied method

China's written historical record dates back 3000 years. The statistical records used in our study include data from southeastern coastal China and Taiwan (Fig.1). The data source upon which our study is based a book titled: A Syllogism of China's "Meteorological Record over the past 3000 Years" (Zhang, 2013). This book consists of 7813 pieces of documentary evidence from China's historical documents including 7713 pieces from local government bodies and another 28 from other historical documents. In total, there are more than 220,000 recorded events. After thorough verifications of data sources, timing, and event locations found in the record primary source reports were kept and duplicates eliminated. This is, by far, the most complete and commonly accepted climate record from China's documented history.

Considering the evolution of typhoon-related keywords over the years, besides using the specific keywords " Typhoon" and "Jufeng" to search for records since 1000 A.D. Related expressions such as "strong wind" (大風), "rainstorm" (暴雨), and "storm surge" (風暴潮) were also applied to our search. However, the terms jufeng and typhoon rarely appeared in the historical record prior to 1000 B.P. So, for this earlier period, we added additional terms that are possibly associated with "typhoon" such as "trunk pulling" (拔木), "tree pulling" (拔樹), "collapsed building" (覆屋), and "wind storm" (暴風) to our statistical study. We attempt to reconstruct the time of occurrence and the location of paleotyphoons along the coastal region in China, and to understand the evolution of typhoon development over a long period of time. It is worth to note that every episode would be recorded in historical documents due to a significant damage or a disaster. As a result, we speculate that the strengths of typhoons would be above moderate. All ancient Chinese literatures were listed in the appendix of Liu (2015). Table 1 shows some illustrations of original historical source.

Table 1. Illustrative quotations from selected historical sources in China.

| Occurring time | Descriptions | Locality | Data source |
|---|---|---|---|
| 798 A.D. August | Strong wind destroyed the buildings and overturned the boats. | Guangdong | The New Book of Tang , The notes of the Five Elements |
| 1380 A.D. September | Jufeng and heavy rainfall damaged the woods and houses. Many people died in this disaster. | Fujian | Ming Taizu (The first founder of the Ming Dynasty) Memoirs, Volume 133 |
| 1673 A.D. August | Jufeng and heavy rainfall happened. The roofs were thrown up and tall trees were snapped off. | Guangdong | Qing Qianlong Years, Chaozhou Prefecture Records, Volume 11, The Disastrous and Fortunate Events |
| 1750 A.D. August | Strong jufeng destroyed the buildings and the surge smashed several hundreds of merchant ship. | Taiwan | Qing Jiaqing Years, Updated Taiwan County Records, Volume 5, The Fortunate and Abnormal Events. |

| 1831 A.D. July | Jufeng and heavy rainfall caused flooding and seawater intrusion in the coastal range. More than 9500 people died and the houses floated away in flood. | Shanghai | Qing Guangxu Years, Chongming County Records, Volume 5, The Fortunate and Abnormal Events. |
| --- | --- | --- | --- |

## 4. Results

### 4.1 Statistical results on the frequency of typhoon landfall

The statistical data collected for the southeastern coastal regions of China includes data for: Hainan, Guangdong, Fujian, Taiwan, Zhejiang, Shanghai, Jiangsu, and Shandong (Fig. 1). When we categorized typhoon landfall locations based on latitudes, Fujian and Taiwan are recognized as one region due to their similarities in latitude and the same as Jiangsu and Shanghai. It is notable that prior to 2000 years BP the historical record of China lacks data of typhoon activity. Consequently, this study focuses on data collected over the past 2000 years. Furthermore, data for the period 1945-2013 A.D. were collected from the northwestern Pacific Ocean TC records established by the Joint Typhoon Warning Center (JTWC). The statistical results were divided into three different time frames based on keyword results and database sources: (1) 0-1000 A.D.; (2) 1000-1910 A.D.; and (3) 1945-2013 A.D. To plot the number of typhoons occurring as a function of time, typhoon events in any given decade were collectively plotted to create an interdecadal bar-graph dating from 1000 AD to the present (Fig. 2). The number of events which occurred in any given decade relates closely to the age of historical documents and how well they have been preserved. Records relating to TC landfall between 1945-2013 A.D. is reliant on satellite acquired data meaning the data source is highly reliable in terms of its location and intensity. Consequently, Figure 2 shows extreme growth in the number of recorded TCs in the latter years of the twentieth century. Moreover, Liu et al (2017) published TC landfall data for the northwestern Pacific Ocean region during 1945-2013 A.D. which corresponds to the results seen here. The figure 2 shows clearly that TC activity grew extraordinarily at around 1500 A.D. and has persisted to the present.

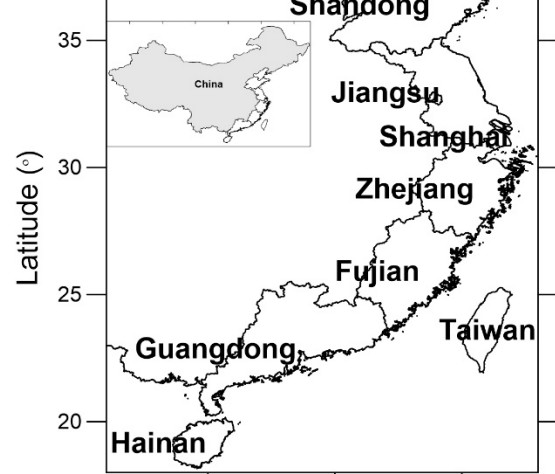

Fig. 1 Southeastern coastal regions of China and Taiwan

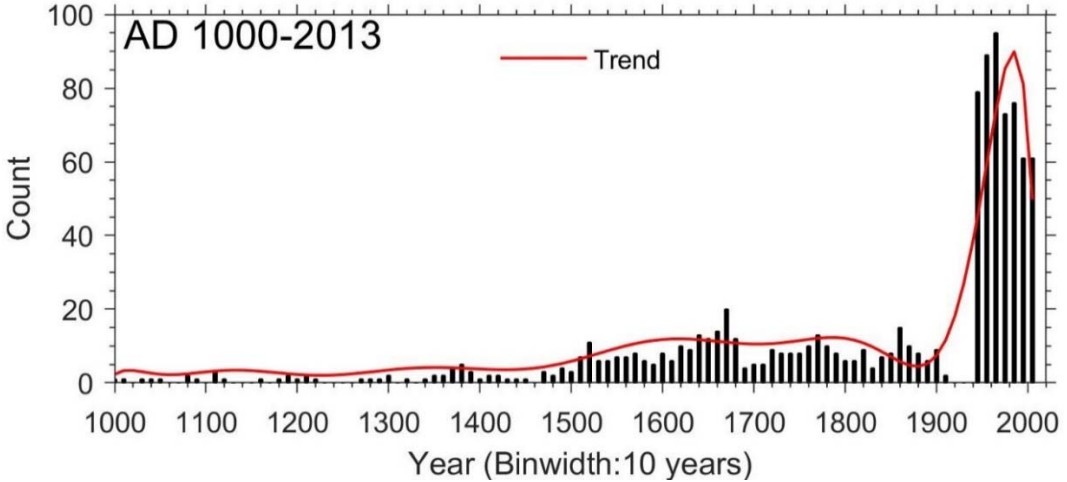

Figure 2. Historical paleotyphoon data compiled over the past 1000 years from China's
166             historical record and JTWC data for southeastern China and Taiwan. Each bar
167             in the bar-graph represents the collective number of typhoons occurring in any
168             given decade.

### 4.1.1 Statistic typhoons during 0-1000 A.D.


The term "Jufeng" did not appear in any historical documents before 1000 A.D.
Some of the documents, however, only mentioned disaster conditions such as "trunk
pulling", "tree pulling", "collapsed building", "wind storm" and "torrential rain". Given
these limitations, all the typhoon records from 0-1000 A.D. were examined for using
these assemblage proxies. The original results are listed in Table 1 of the supplementary
file. There were 124 possible typhoon events found in the records, which have been
presented in Figure 3. The figure shows for the time period 0-1000 A.D., there were on
average 1.2 typhoons recorded every 10 years. Based on this result, we define the
periods that average more than 1.2 typhoons each 10 years plus recorded continuously
50 years as a high frequency typhoon period. Figure 3 shows that the periods 490-510
A.D. (South-North Dynasty) and 700-850 A.D. (Tang Dynasty) were periods of
frequent TC invasions. Our statistic results respond that why many storm damages were
mentioned in ancient poetries during the Tang Dynasty (Louie and Liu, 2003).

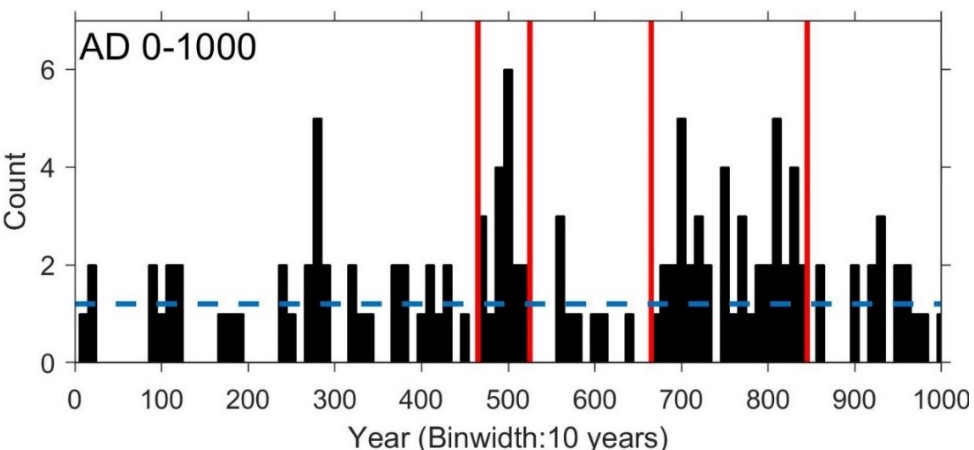

Fig. 3 Statistics showing the number of typhoons during 0-1000 A.D. The red range
means high frequent periods of TCs.
**4.1.2 Statistic typhoons during 1000-1910 A.D.**

Figure 4 gives a total of 408 events relating to the terms "Jufeng" and "Typhoon"
for the period 1000-1910 A.D. Original data are listed in Table 2 of the supplementary
file. Starting from 1460 A.D., TC landfall by suddenly number started to increase
peaking between 1670-1679 A.D. Other periods with substantial numbers of TC
making landfall are: 1520-1529 A.D, 1770-1779 A.D, and 1860-1869 A.D.. During
these times, recorded typhoon landfall was greatest in the Guangdong region (Fig. 6).
To make sure the historical record accurately reflected climatic conditions for the
period examined, a search of the record was conducted for anomalous climatic events
such as flooding, snow storms, and droughts and so on. It was found that there were
extensive gaps in the data for the periods 1270-1320 A.D. and 1400-1450 A.D.. The
two periods that corresponded to the advent of the Yuan and Ming Dynasties,
respectively. All original data sources are listed in Table 5 of the supplementary file.
The Yuan Dynasty was established by foreign-led dynasty of Kublai Khan of Mongolia.
It was a period described by much internal strife and rebellion. The lack of good climate
data in the historical record for the period 1400-1450 A.D. at first glance might seem
surprising as it is the time of the Yongle Emperor and the promotion of Admiral
Zhenghe, the eunuch commander of the 7 great international tributary voyages across
the South China Sea and Indian Oceans (1405-1430 A.D.). It would seem likely that
weather conditions, especially TC would be of great import to China and this
information would have been carefully recorded. This period is well described in the
book: 1421 (Menzies, 2008). In fact it is thought Zhenghe did record such detail, but
much of it was lost or burned during "Eunuch Conflict" and much internal conflict at
the death of Emperor Yongle. The historical records were terminated in AD 1911
because the Qing Dynasty was overthrown and a civil war was fought in China for a
long period of time. In addition, the World War I happened during 1914-1918 A.D. and
the World War II took place during 1939-1945 A.D. Therefore, China lacks climate
records in the turmoil of war during this period in history.

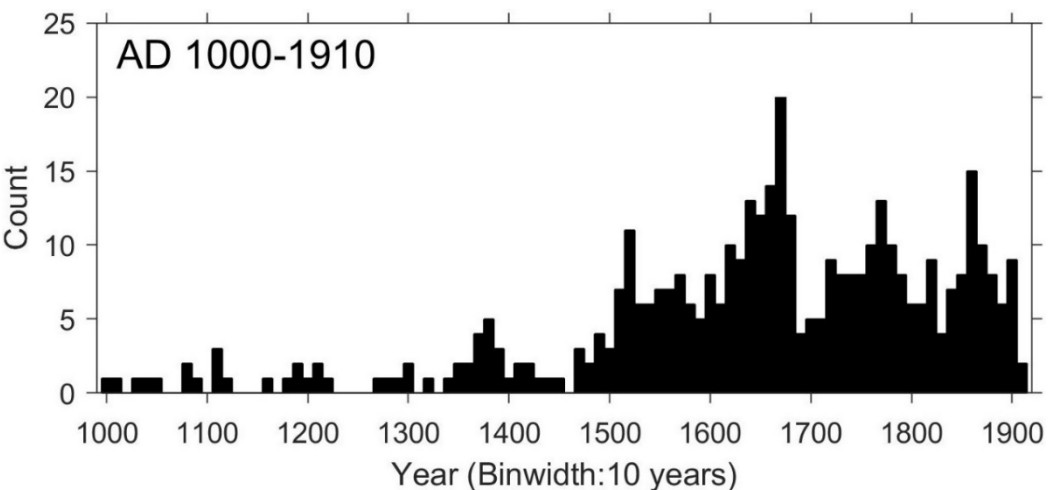

Fig. 4 The numbers of typhoons occurring per decade for the period 1000-1910 A.D.

**4.2 The change in months of the year when typhoons occur**

To further investigate any changes in the timing of TC landfall occurrence

annually, TC landfall data was collected and analyzed for the three different time periods: 0-1000 A.D.; 1000-1910 A.D.; and 1945-2013. The results are shown in Figure 5 and monthly statistics listed in Table 3 of the supplementary file. Before 1000 A.D., TCs in China mostly occurred in June, July, and August (Fig. 5a). However, after 1000 A.D., the entire trend in arrival times shifted by one month with TC landfall occurring predominantly in July, August, and September (Fig. 5b). The majority of statistics after 1000 A.D. were collected during the LIA (1400-1850 A.D.). Figure 5c shows statistics for the period 1945-2013 A.D. The timing of recent TCs making landfall in southeastern China is quite similar to that which occurred during the LIA period. Recent data shows that TC occurrence in the entire northwestern Pacific Ocean region can last until as late as October, November, and December with TCs making landfall in Vietnam, Philippines, and Thailand after September (Liu et al., 2017). It is assumed that this relates to seasonal changes in the positions of the subtropical high and ITCZ of the northwestern Pacific Ocean region. The ITCZ begins migrating north away from the equator in March or April. It reaches its northernmost position in August, before migrating south in September (Waliser and Gautier, 1993). The question this study raises is what happened to shift the predominant timing of TC arrival in southeastern China from between June~August during 0-1000 A.D. to between July~September after 1000 A.D. One likely explanation is the ITCZ being at a higher latitude before 1000 A.D. (Rehfeld et al., 2013), resulting in earlier (June~August) TC formation.

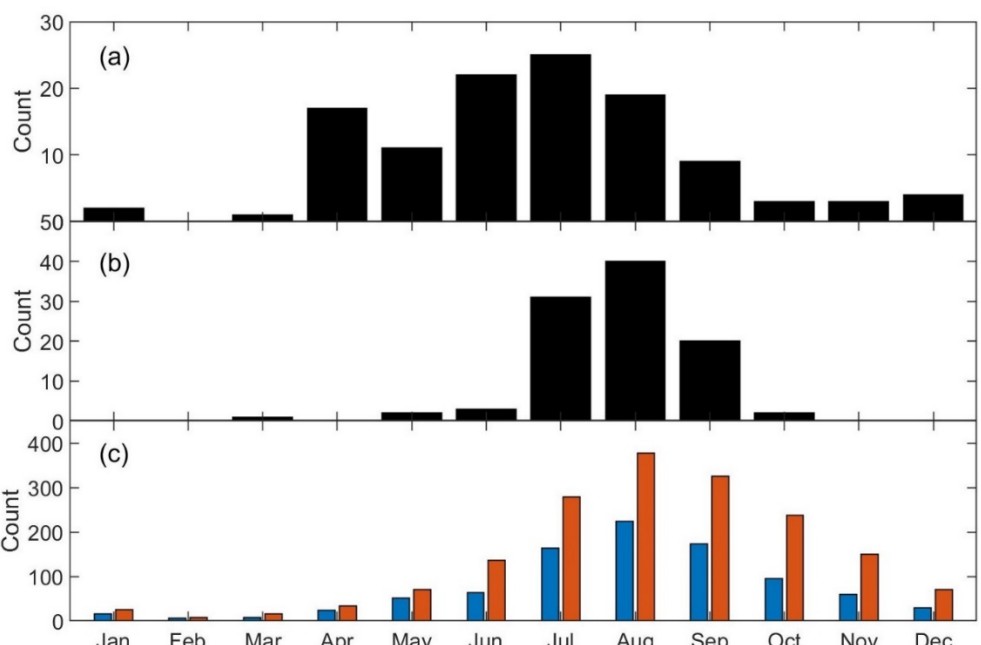

Fig. 5 Statistics on TCs that struck China (a) 0-1000 A.D. (b) 1000-1910 A.D. (c) 1945-2013 A.D. Blue bars indicate the ones that hit China; the red bars indicate the ones that hit the north-western Pacific Ocean region.

**4.3 The spatial distribution of the typhoons - the relationship between landfall locations and occurrence frequencies**

Not all historical record gave detail on where TCs struck until 1000 A.D.; therefore, this study focuses solely on the landfall locations of paleotyphoons between 1000 and 1910 A.D. The number of typhoons that struck each providence in China are shown in Figure 6. Table 4 of the supplementary file gives additional detail on landfall

locations. For the period 1000-1910 A.D., Guangdong was struck by the most TCs. On
the whole, the number of TCs making landfall increased dramatically after 1500 A.D.
with the number of typhoons hitting Guangdong peaking between 1660-1680 A.D. By
contrast, regions north of Fujian did not record any increase in typhoon activity during
this time-period. The number of typhoons striking Zhejiang and Jiangsu, however, did
start to increase after 1700 A.D.
Newton et al. (2006) proved that the warmest temperatures in the Indo-Pacific
Warm Pool occurred during the Medieval Warm Period while the coolest
temperatures occurred during the Little Ice Age. In particular, the lowest temperatures
occurred around 1660-1680 A.D. within the period of the Maunder Minimum (1645-
1715 A.D.). Therefore, it is thought that the sudden change TC tracks around 1700
A.D. may relate to a change in temperature lows in the northern hemisphere and a
shift in the location of the ITCZ.

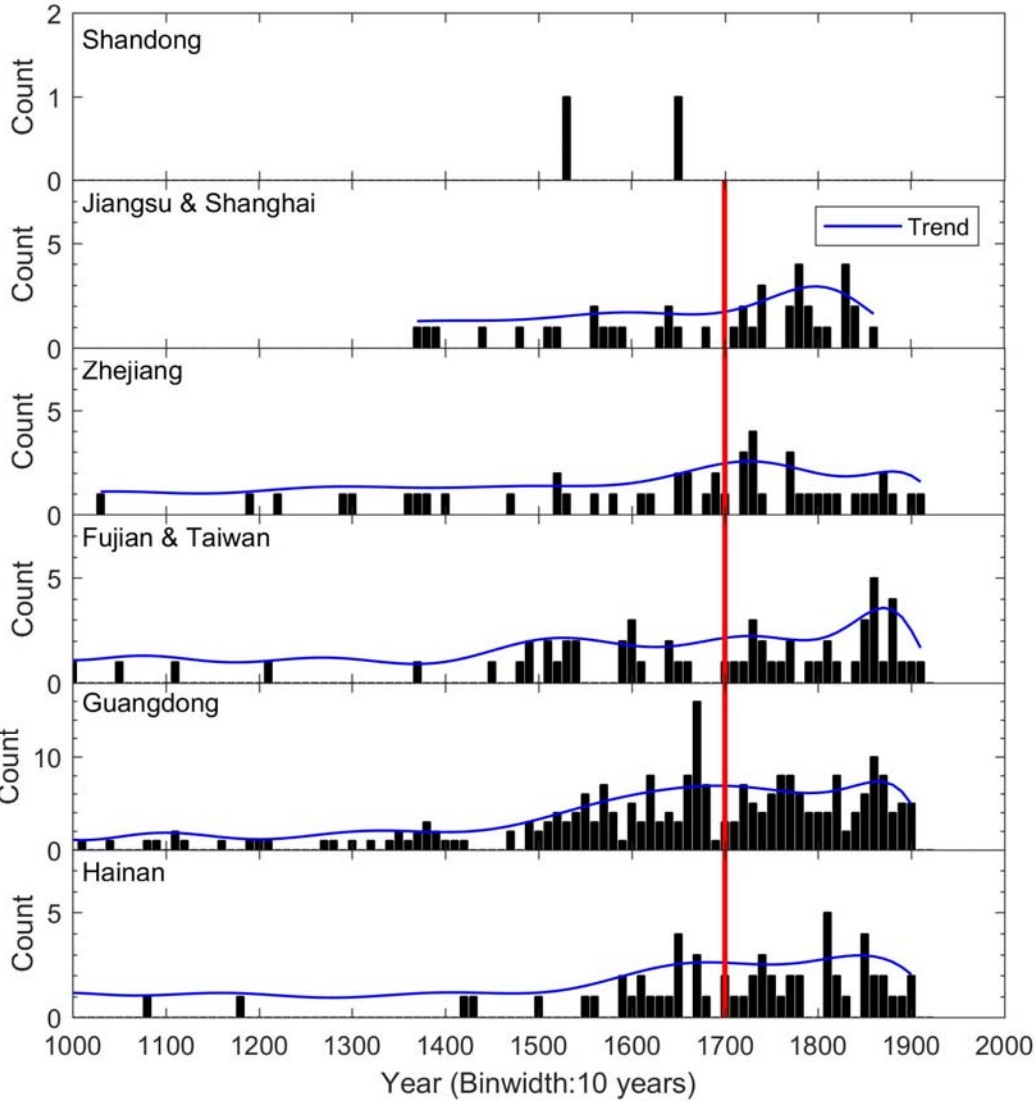

Fig. 6 The number of typhoons that struck the southeastern regions of China and
Taiwan during 1000-1910 A.D. (Red line means the time boundary of 1700 A.D.
More TC made landfall in Guandong before this time, but more TC made landfall to
northward after this time)

## 5. Discussions

### 5.1 Northwestern Pacific Ocean paleotyphoon track changes during the MWP and LIA

Conserving historical documents has always been a difficult task. Racial conflicts, war, rebellion, and inter-court feuds could all result in precious data being damaged, destroyed or lost during certain periods in history. Consequently, statistics on paleotyphoons recorded in the historical record are only semi-quantitative. On the other hand, they are very useful in terms of noting the location of landfalls and the precise timing of such events. To help overcome any anomalies in the typhoon record lost to documented history and avoid any confusion regarding the intensity of events, this study also looked at the geological record of paleotyphoons derived from lake sediments in northeastern Taiwan (Chen et al., 2012; Yang et al., 2014; Wang et al., 2013, 2014, 2015). Since the topography of northeastern Taiwan's Yilan region is quite unique with the summer monsoon being blocked by mountains and rainfall being mainly supplied by the winter monsoon and typhoons (Chen et al., 2012), the region is very helpful for studying TCs tracking in the Northwestern Pacific. In fact, large-scale river terraces have occurred due to typhoon rainfall and this record in preserved in the mountain areas of Yilan since 2.7 ka BP (Hsieh, 2017).

In order to correlate the number of paleotyphoons from historical data with the geological record of lake sediments, the Southern Oscillation Index (SOI), intensity of paleotyphoons determined from sedimentary particle size at Taiwan's Lake Dahu, and paleotyphoon signals from lagoon sediments in Kyushu, Japan (Fig. 7) are referenced and compared. Results suggest that typhoons struck Taiwan and the southeastern coastal region of China mostly during La Nina-like stages (Figs. 7a, b, c) (Chen et al., 2012). This outcome matches that mentioned by historical maritime disaster events caused by paleotyphoons in the last 1000 years in Liu et al. (2017). According to Liang and Zhang (2007), the chances of a typhoon making landfall in the southeastern coastal region of China during La Nina years is higher than that during El Nino years. If we started entering an El Nino like stage after 1900 A.D., this means the number of typhoons striking Japan in the future will very likely increase compared to what we see now. This trend in the data since 1700 A.D. shows a gradual increase in typhoon numbers moving north and away from Guangdong (Fig. 6). It has also been shown that the number and intensity of typhoons recorded in Taiwan's lake sediments has grown since the LIA (1400 A.D.) which seems to match the general trend in the recorded number of historical events pretty well (Fig. 7a and c). This period also coincided with the timing of flooding events in southern China (Fig. 7d). Park et al. (2017) investigated the records of lake sediments in the East Asia region. Their study noted that along coastal regions including Jeju Island (Korea), lakes in Yilan (Taiwan), Lake Huguangyan in Guangdong, and lakes on Hainan Island relatively drier conditions prevailed during MWP and wetter conditions during the LIA. This may be due to an increase in rainfall caused by typhoons along the coast.

This study, therefore finds that the northward migration of the ITCZ during the MWP caused typhoons to move north toward Japan. In contrast, typhoons moved toward southern China during the LIA due to the southward transition of the ITCZ. This seems to be a reasonable explanation and is not out of step with other regional studies (Rehfeld et al., 2013; Chen et al., 2015; Xu et al., 2016).


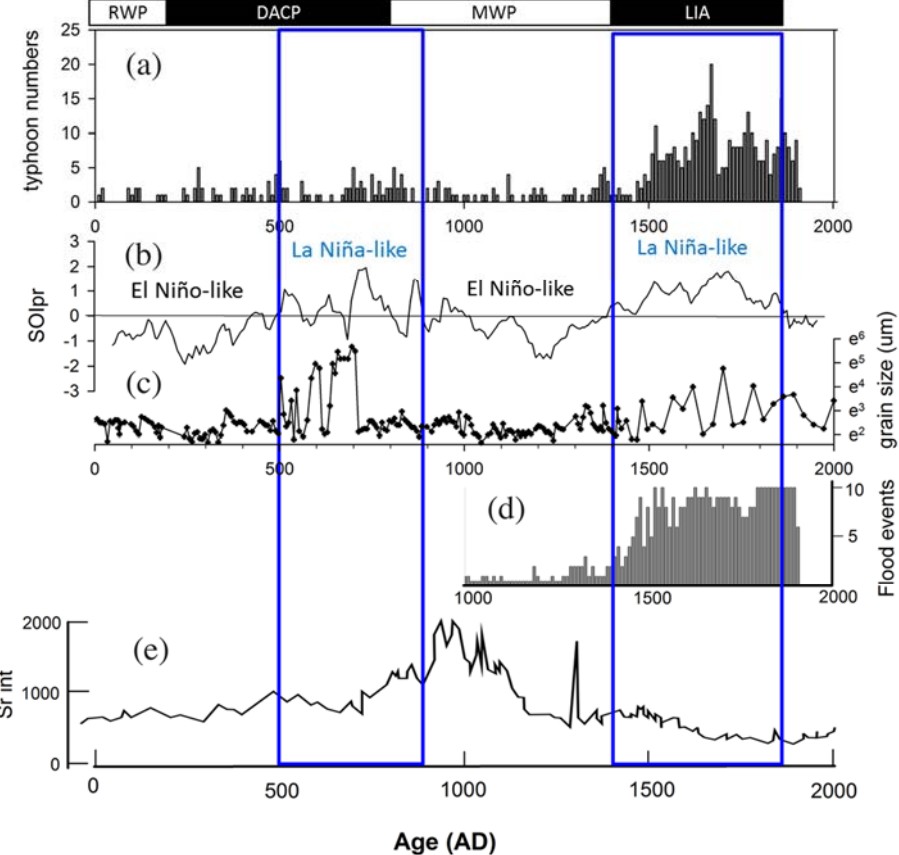

Fig.7 Correlations between typhoon events and ENSO. (a) Number of typhoons
recorded in Chinese historical documents for the last 2000 years. (b) SOI (Yan et al.,
2011). (c) The change in particle sizes from lake sediments from Yilan, Taiwan
indicating the change in magnitude of typhoon rainfall (Chen et al., 2012). (d) Number
of flooding events recorded in Chinese historical documents (Chu et al., 2002). (e)
Variation of Sr in lagoon sediments from Kyushu, Japan indicating influences from
super strong typhoons (Woodruff et al., 2009).

**5.2 The linkage between ancient TCs of the northern Atlantic Ocean and**
**northern Pacific Ocean**

Donnelly and Woodruff (2007) first suggested that the number of hurricanes in
the Caribbean area has been increasing over the last 4000 years. According to ancient
hurricane research along the Gulf Cost, Caribbean Sea to Puerto Rico, hurricane tracks
show an antiphase in time series data (McCloskey and Liu, 2012, 2013; McCloskey et
al., 2013; Liu et al., 2015). During the MWP, more TCs made landfall in the Gulf Coast
as the strength of the Bermuda High enhanced and the ITCZ moved northward. During
the LIA, more TC made landfall on the Caribbean Sea (McCloskey and Knowles, 2009;
McCloskey and Liu, 2012, 2013; McCloskey et al., 2013). In 1650 A.D., TC frequency
reached a peak, and after 1850 A.D. TCs began to move toward Florida and Bermuda
with the northward movement of the ITCZ (Baldini et al., 2016). Ancient lake sediment
data from Yilan, Taiwan reveals the period in history when paleotyphoons occurred
most frequently. This timing highly correlates to the time of paleohurricanes recorded
in Belize (McCloskey and Liu, 2013). This suggests that the migration paths of TCs in

both the northwestern Pacific Ocean region and the northwestern Atlantic Ocean region are closely related. The TC activity happened during 200–600 yr BP and 1450–2600 yr BP in Belize, and it occurred during 200-500 yr BP, 1300-1500 yr BP and 2000-2300 yr BP in Taiwan's lakes (Chen et al., 2012). This phenomenon indicates a close association between TC activity in the North Pacific Ocean and the North Atlantic Ocean.

**5.3 The Track of TCs corresponding to the NAO during the LIA**

Since the ITCZ and Westerlies both link to the Hadley Cell, and the position of middle-latitude storms are dragged by the westerlies which is influenced by the North Atlantic Oscillation (NAO) (Hurrell, 1995; Morley et al., 2014), we compared the NAO record with the track of TCs. In order to compare our tracks of TCs with the NAO, we created an index of TTC1 to represent the track of TCs that either move toward southern China or toward northern China (TTC1 = $\sum X_i F_i$ ). $X_i$ is the number of typhoons that had made landfall in that certain province, and $F_i$ means the location factor of the landfall locality (Table 2). When the value of TTC1 is higher, it indicates a larger amount of typhoon landfalls in northern China (Fig. 8). The TTC1 can also be normalized to values between 0~1. Furthermore, we used digitalization to retrieve the average data of 10-year from 2ka NAO index according to the results of Trouet et al. (2009) and Ortega et al. (2015). The results calculated from Trouet et al. (2009) and our TTC1 agree quite well (Fig. 8). However, our records were fragmentary before 1470 A.D. and we lack the historical data from Japan. The results in Figure 8 reveals that our normalized TTC1 corresponding to the $NAO_{touet}$ during the LIA stage, and the 3-point smoothing of the TTC1 shows a very good correlation with the $NAO_{touet}$. This result indicates that the NAO influences the migration of the westerlies and it may also gently affect the tracks of the TCs.

Table 2. Location factor ($F_i$) of various geographical locations in China

| Landfall Locality | Hainan | Guangdong | Fujain & Taiwan | Zhejang | Jiangsu & Shanghai | Shandong |
|---|---|---|---|---|---|---|
| Location factor ($F_i$) | -2 | -1 | 1 | 2 | 3 | 4 |

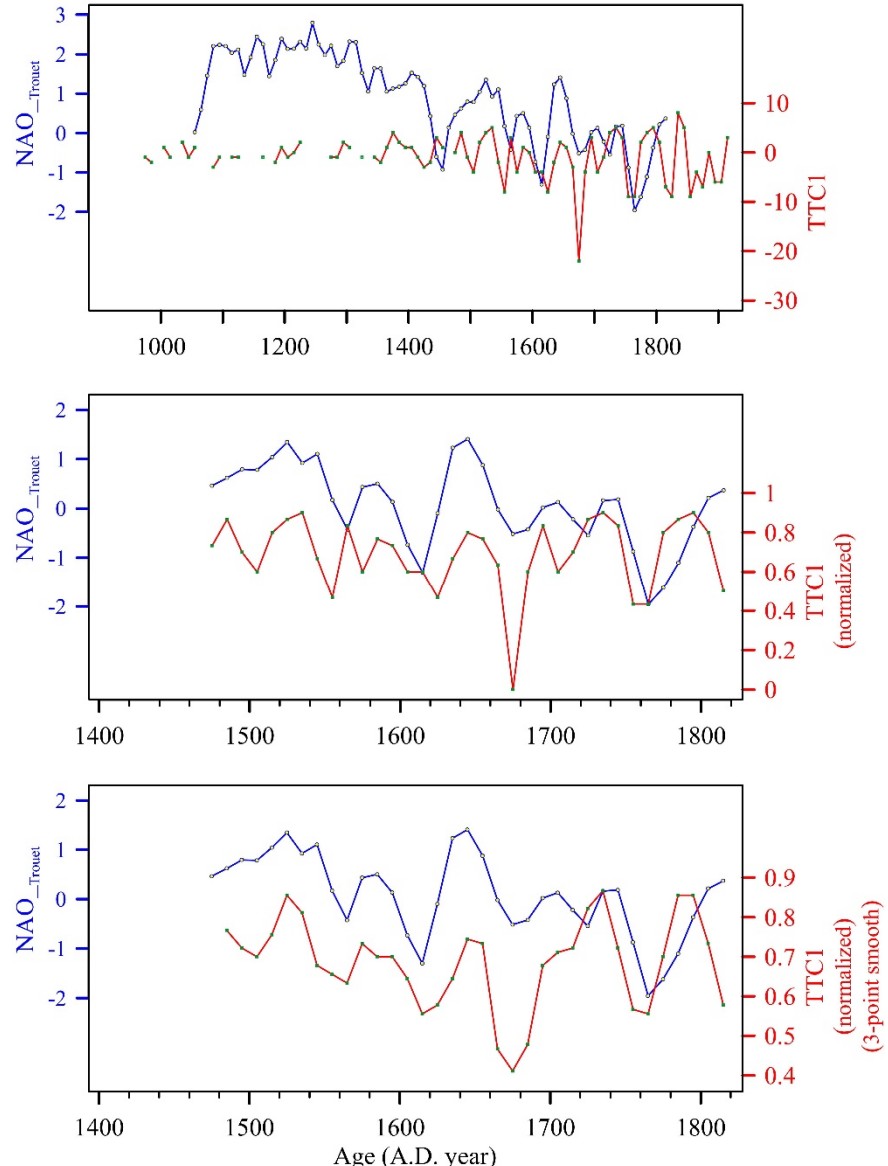

Fig. 8 The relation between the NAO$_{trouet}$ (Trouet et al., 2018) and the TTC1

387        After we performed the wavelet analysis, we found that the TTC1 shows both 30-
35 yr and 55-65 yr cycles during the LIA stage (Fig. 9). This result is also consistent
with the frequency of typhoon landfall over Guangdong Province of China during the
period of 1470 A.D.~ 1931 A.D. based on a different data source (Chan and Shi, 2000).
The 60 yr cycle is clearly present in the Pacific Decadal Oscillation (PDO) and the
Atlantic Multi-decadal Oscillation (AMO), with phases coherent with a planetary signal
since at least 1650 A.D. to 1850A.D. (Scafeta, 2012; Solheim, 2013). This implies that
the PDO also affects the TTC1 cycle.

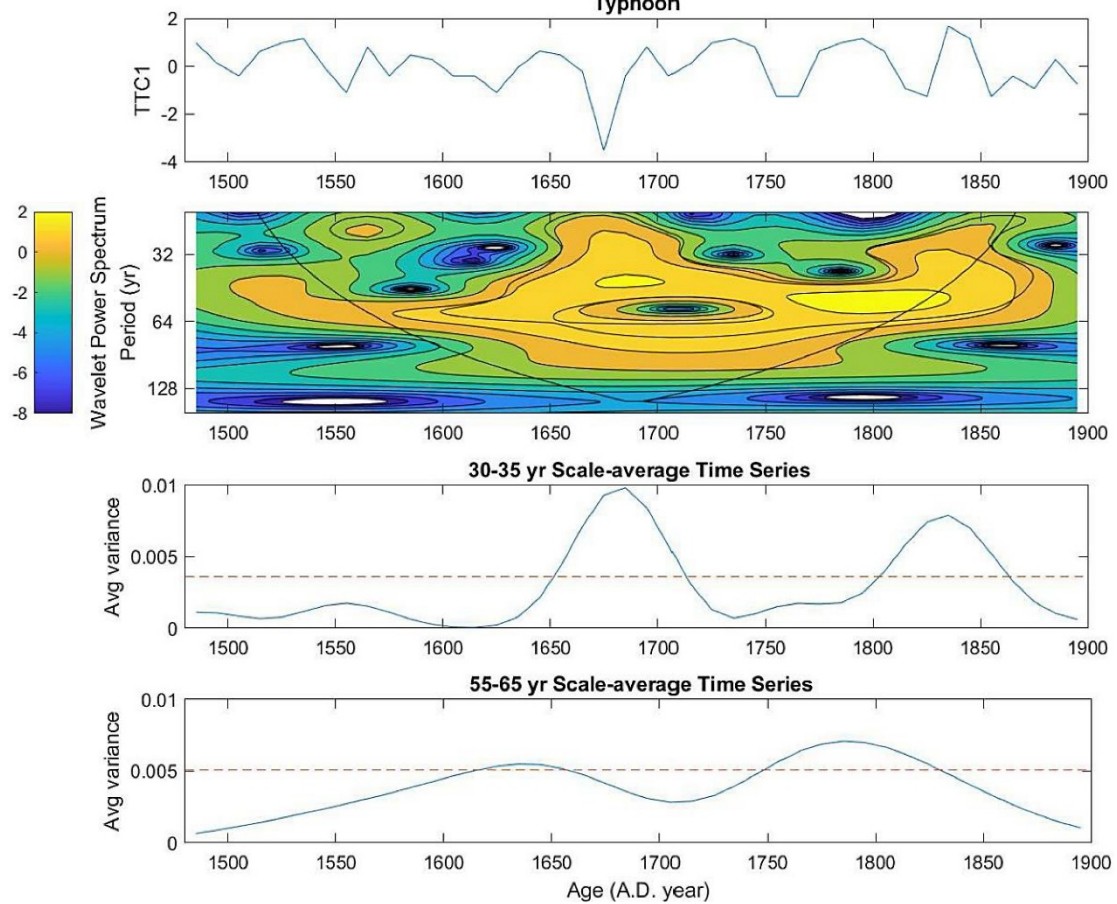

Figure 9. The wavelet analysis of the TTC1 during the LIA

## 6. Conclusions

We statistically analyzed Chinese historical documents to understand the relationship between the MWP, LIA and movements in the ITCZ. Our conclusions are very similar to those found in previous studies, indicating that China's documented historical record is an invaluable asset in the study of climatological phenomena. The conclusions are as follows:

(1) Before 1000 A.D., TCs struck China mostly in June, July, and August. The timing of TC landfall shifted to July, August, and September after 1000 A.D.

(2) Statistical analyses of China's historical documents show that there was a sudden increase in the frequency of paleotyphoons in 490-510 A.D., 700-850 A.D. and since the beginning of the LIA (1400 A.D.).

(3) Correlating lake core records from Taiwan and Japan proved that more typhoons made landfall in Guangdong and Taiwan during the LIA.; whereas, more typhoons made landfall in Japan during the MWP.

(4) Most typhoons made landfall in Guangdong at the coldest era of LIA. Typhoon tracks started migrating towards Fujian and farther north after 1700 A.D., indicating that there is a northward trend in typhoons towards Japan.

(5) The track of TCs has 30-35 yr and 55-65 yr cycles during the LIA stage, the result is consistent with the variation of the NAO and the PDO cycles.

Paleoclimate research covering the last 2000 years since the late Holocene mainly focuses on three drastic temperature fluctuation periods, including the MWP, LIA, and

the global warming of the past 200 years. Our study shows that the paths of paleotyphoons between the MWP and LIA closely related to the migration of the ITCZ. The results also demonstrate that the migration paths of TCs in the northern Pacific Ocean and the northern Atlantic Ocean are highly correlated with the NAO and the PDO cycles.

## Acknowledgements

This study was supported by the National Taiwan Ocean University and grants NSC103-2116-M-019-003 and NSC106-2116-M-019-004 from the National Science Council of Taiwan. We are grateful for Prof. Kam-Biu Liu at Louisiana State University, who started the research of paleotyphoons by using historical records. His research was greatly edifying.

## Appendix A. Supplementary data

Supplementary data related to this article can be found at xxx.

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
