# Peer review of "China's historical record in the search of tropical cyclones corresponding to"

_Climate of the Past, 2018_

## Referee Comment (RC1) · J. Elsner (Referee) · 7 Aug 2018

This manuscript uses historical records to reconstruct tropical cyclones in China over the past two thousand years. The authors utilized keyword searches to find mentions of cyclones, using period specific terms. The search was done on the data in "A Syllogism of China's "Meteorological Record over the past 3000 Years". The authors show how these weather systems varied over the past two thousand years, and how it related to other major climate trends like the Southern Oscillation, Medieval Warm Period, and the Little Ice Age.

The manuscript does a few important things. It adds to the growing climatology literature that combines paleo historical methodologies. It does an excellent job demonstrating how long of a period can be covered by applying creative and novel methods to historical records. It also brings attention to the fact that appropriate records exist for a wide range of locations and periods.

The manuscript could be greatly enhanced by a deeper discussion of the data. An extra paragraph or two in section 2: "Paleo typhoon records from China's official historical documents" that discussed "A Syllogism of China's "Meteorological Record over the past 3000 Years" (Zhang, 2013) would be ideal. This could discuss the original data, how it was digitized, and its limitations. I was not able to find any other references to it in english. If it is possible to access this data, information how to do so would be very useful. Since it is such a rich and important source of data, a discussion here would be valuable to many readers. This will also help show that the authors did their due diligence, even though there does not seem to be a historian on their team.

Most of what is presented here amounts to histograms and frequency counts. This seems like the logical first phase in analyzing the data. However, if there is future work planned with this data, that could be made more clear. Alternatively, the data could be used with another data set to validate that being used. There are many ways this could be done. For instance, correlating or regressing the data with the SOI data used in Figure 7 would probably be sufficient. This could produce confidence intervals to show the reliability of the data.

Minor notes:

"Syllogism" in Zhang 2013 is misspelled in the references.

A reference to figure 1 on line 109 would be useful.

The map in figure 1 is very difficult to read and is missing basic cartographic elements. I would suggest zooming into the region, adding a regional map in the northwest corner, changing the fill color of the provinces to white, and changing the text color to black.

It might be useful to run the coastline shapefile through a simplifying or smoothing algorithm to get rid if the thicker black lines there.

In figure 6: it is not clear in the caption what the red line represents. This could be added to the legend with the trend line.

In figure 7: extend the axis line of (d) to match the other elements

This review was done by my graduate student Greg Burris.

---

## Author Comment (AC1) · 24 Aug 2018

Dear Prof. Elsner,

We very appreciate your comments and so quick response. We will do all revisions in the next version of manuscript. However, we don't extend the axis of the age to 0-1000 A.D. in figure 7(d), because the original data of Chu et al. (2002) only show the data between 1000-1910A.D. If we extended the axis, readers would misunderstand no events occurring at 0-1000 A.D. We will add a paragraph in section 2 to describe our method for more detailed as bellow. It is really important for our keywords in this research. Many thanks for your attentions.

[Figure]

Considering the evolution of typhoon-related keywords over the years, besides using the specific keywords " Typhoon" and "Jufeng" to search for records since 1000 A.D. Related expressions such as "strong wind", "rainstorm", and "storm surge" were also applied to our search. However, the terms jufeng and typhoon rarely appeared in the historical record prior to 1000 B.P. So, for this earlier period, we added additional terms that are possibly associated with "typhoon" such as "trunk pulling", "tree pulling", "collapsed building", and "wind storm" to our statistical study. We attempt to reconstruct the time of occurrence and the location of paleotyphoons along the coastal region in China, and to understand the evolution of typhoon development over a long period of time. It is worth to note that every episode would be recorded in historical documents due to a significant damage or a disaster. As a result, we speculate that the strengths of typhoons would be above moderate. All ancient Chinese literatures were listed in the appendix of Liu (2015).

Corresponding author Huei-Fen Chen
* * *
[Figure]

**Fig. 1.**

---

## Referee Comment (RC2) · Anonymous Referee #2 · 19 Sep 2018

Overall, this is a very important topic on paleotyphoon patterns in China. The findings should have both academic implication and practical value to address the potential risks of extreme weather in the future. However, the manuscript itself should be revised substantially according to current form.

First, the descriptive analysis on paleotyphoon is important. But, this is the first practice so far. The authors should justify the innovative points of their works. Especially, the data is from Zhang (2013), which is not a first-hand data.

Second, why author compared with ENSO? Why authors chose these ENSO data? The comparative analysis between ENSO and paleotyphoon is so rough without any

calculation. The authors should have done some statistical analysis because this is the research in physical sciences.

Third, in terms of NAO or Pacific Oscillation, the authors did not use any data series at all. They only discuss the linkage qualitatively. I think it would be better to use some reconstructed data series of NAO or Pacific Oscillation to compare with paleotyphoon.

Fourth, since the last phase of the research 1945-2013 could use the satellite data, could the authors also use different satellite data, like ENSO to support their conclusions?

These are some major points I found during my review. I hope it could help the authors to improve their work.

---

## Author Comment (AC2) · 2 Oct 2018

Dear Editor,

We appreciate the reviewer give us very important suggestions. We will revise his comments in our manuscript, but we need time to do the statistical analyses before the end of October. The follows are our recent quickly response according his questions.

(1) The data of Zhang (2013) is not a first-hand data. Before, we asked Dr. Zhang and she answered that they had more data, but many of unconfirmed records they never published. So, we only can use the published book. The data has higher credibility

because they deleted repeat data from different documents. (2) This ENSO data was calculated from many cores of precipitation proxies in western and eastern equatorial Pacific Ocean. It means the "Southern Oscillation Index" of the Pacific Oscillation. Although it is not a temperature proxy of Pacific Ocean but it shows a good correlation with our time interval just covers the past 2,000 years. When we want to find "the same time scale" and a suitable data "resolution of time", we find that this data is more suitable for our correlation. (3) This suggestion is very good and useful for us. Therefore, we will find a suitable NAO index and compare that data with Southern Oscillation index and our paleotyphoon data. We will add a plot in the revised manuscript and try to use statistical analyses to dissect the relation coefficient between these two physical parameters. (4) The relation of recent typhoon and ENSO had been discussed in many atmospheric research before (Ho et al., 2004; Wu and Lau, 1992; Lander, 1993, 1994; Chan,1985, 2000; Lander, 1994; Elsner and Liu, 2003). Their researches had more statistical methods, but they all used only short time scale. Modern discovery recognized the occurrence place of warm pool as linkage to ENSO and influenced the occurrences of typhoon. Therefore, we do not do the same thing again.
* * *

---

## Author Comment (AC4) · 12 Oct 2018

Dear reviewer,

We have tried to do our best for collecting the raw data of SOI index (Yan et al, 2011), and NAO data during 2ka. We find that only has a little positive relation coefficient (R2=0.156) between the TC number of south China and La Nina-like stage, because we lack the historical data from Japan and Philippines. In this study we only can get the historical data from China. In order to collect the same time range of NAO index, we found two references as follows (Ortega et al., 2015; Trouet et al., 2009) but we cannot get raw data from the authors. We had tried email to them, but no response for

us. Therefore, we only can use digital plotting to get the possible data, and we find all data are not linear time interval. If we want to compare our TTC2 (the northward and southward of TC track ) with the NAO, we need get the same time interval. However, the time in our research is correct while that of SOI and NAO are simulated from many core data. The dating of C14 has it problem and the time shift may happen in the core data. So, we only can get preliminary pattern of the new figure. In our data, also has some problem that some period lacks historical record, so you will find that some TTC2 proxy lack data before AD1450, if we have more TC from Japan the result should be very different. We think this need more work to combine the TC data from other places to finalize the track relation with NAO. We will add this in our discussion section. Do you think this should be added in supplement file or in the main text?

Figure explanation: (1) What is the TTC2? It was calculated from the (xi)/total (xi)*(position factor). Xi means the number of TC in that province and total (xi) means total number of TC in South China. Position factor from the south to the north province is -2, -1, 0, 1 ,2 ,3 ,4. So, when the larger TTC2 value means the TC moving to northward. (2) The SOI was recalculated from Yan et al. (2011) for an average of every 10-year (3) The NAO Trouet was recalculated from Trouet et al. (2009) for an average of every 10-year (4) The NAO mc was recalculated from Ortefa et al. (2015) for an average of every 10-year

References: Ortega et al. (2015) A model-tested North Atlantic Oscillation reconstruction for the past millennium. Nature 532, 71-74, doi:10.1038/nature14518.

Trouet et al. (2009) Persistent Positive North Atlantic Oscillation Mode Dominated the Medieval Climate Anomaly. Science 324, 78-80.
* * *
**Fig. 1.** Tracks of TC (TTC2) compared to SOI and NAO phase

---

## Author Comment (AC5) · 22 Oct 2018

Dear referee #2 and editor,

We have finished the data analysis of NAO and wavelet analysis of our data in LIA. We want to add the section 4.3 for discussing more. We will revise in our manuscript and add table 1, figure 8 and figure 9.

4.3 Track of TCs corresponding to the NAO during LIA

Since the ITCZ and Westerlies both link to the Hadley Cell, and the position of middle-latitude storms are dragged by the westerlies which is influenced by the North Atlantic

[Figure]

Oscillation (NAO)(Hurrell, 1995; Morley et al., 2014), we try to compare the NAO with the track of TCs. In order to compare our tracks of TCs with NAO, we created an index of TTC1 which means the track of TCs moving to southern China or northern China (TTC1 = ïČěXiFi ). Xi means the numbers of typhoon landfall in that provinces, and Fi means the location factor of landfall locality (Table 1). When the value of TTC1 is higher, it indicates more typhoon landfall in northern China (Fig. 8). The TTC1 also can be normalized to 0∼1. Furthermore, we used digitalization to get the average data of 10-year from 2 ka NAO index according to the results of Trouet et al. (2009) and Ortega et al. (2015). The results calculated from Trouet et al. (2009) and our TTC1 have the same variation (Fig. 8). However, our records were fragmentary before 1470 A.D. and we lacks the historical data from Japan. The results in figure 8 reveals that our normalized TTC1 corresponding to the NAOtouet during LIA stage, and the 3-point smooth of the TTC1 has very good relation with NAOtouet. This result proves that NAO influence the moving of westerlies and gently affect the tracks of TCs.

After we do the wavelet analysis, we found the TTC1 has 30-35yrs and 55-65 yr cycles during LIA stage (Fig. 9). This result is also consistent with the frequency of typhoon landfall over Guangdong Province of China during the period 1470 A.D.∼ 1931 A.D. based on different data source (Chan and Shi, 2000). The 60 yr cycle is clearly present in the Pacific Decadal Oscillation (PDO) and the Atlantic Multi-decadal Oscillation (AMO), with phases coherent with a planetary signal since at least 1650 A.D. to 1850A.D. (Scafeta, 2012; Solheim, 2013). This implies the PDO cycle synchronous with our TTC1 cycle.

References: (1) Chan, J.C.L., Shi, J.E.: Frequency of typhoon landfall over Guangdong Province of China during the period 1470–1931. International Journal of Climatology 20, 183–190, 2000. (2) Hurrell, J. W.: Decadal trends in the North Atlantic Oscillation: regional temperatures and precipitation. Science 269, 676–679, 1995. (3) Morley, A., Rosenthal, Y., DeMenocal, P.: Ocean-atmosphere climate shift during the mid-to-late Holocene transition. Earth and Planetary Science Letters, 388, 18–26, 2014. (4) Ortega. P., Lehner, F. , Swingedouw, D., Masson-Delmotte, V., Raible, C. C. ,Casado, M., Yiou, P.: A model-tested North Atlantic Oscillation reconstruction for the past millennium. Nature 523, 71–74, 2015. (5) Scafetta, N.: A shared frequency set between the historical midlatitude aurora records and the global surface temperature, J. Atmos. Sol.-Terr. Phy. 74, 45–163, 2012. (6) Solheim, J. E.: Signals from the planets, via the Sun to the Earth. Pattern Recogn. Phys. 1, 177–184, 2013. (7) Trouet, V., Esper, J., Graham, N. E., Baker, A., Scourse, J. D., Frank, D. C.: Persistent positive North Atlantic Oscillation mode dominated the medieval climate anomaly. Science 324, 78–80, 2009.

———————————————————

[Figure]

[Figure]

**Fig. 1.** Fig. 8 The relation between the NAOtrouet (Trouet et al., 2018) and TTC1

[Figure]

**Fig. 2.** Fig 9. The wavelet analysis of TTC1 during LIA

---

## Author Response (AR2)

**Dear Editor Prof. Camenisch,**

We very appreciate the two referees for giving us precious comments. We have finalized all corrected point by point. Thank you for accepting our hard works. Our response are listed in this letter.

Referee #1

1. New text has been inserted between lines 52 and 75 in section 1. The new text now disrupts the flow of the introduction. I would urge the authors to review the text for clarity – this is the most important part of the manuscript, and now doesn't really do the study justice.

An: We seriously rewrote the line 52-77 and move a section to line 81-91. We hope our revised for more clarity and sentences smother.

2. Table 1 (page 3) would be better titled 'Illustrative quotations from selected historical sources in China'.
An: Yes, we have done in line 132.

3. Sections 5.2 and the new 5.3 contain passages where the degree of association between variables is perhaps overstated. For example, on lines 351-352, the text reads "This phenomenon indicates a close link between TC activity and…" I would suggest that the word "link" is replaced with "association" to soften the understanding of the relationship. The same occurs in line 372 where the word "proves" is used – this is very strong. You might rephrase the text to read: "This result suggests that the NAO influences…" I would urge the authors to check the remainder of the text for overly strong linkages.
An: Yes, we revised in line 358 and line 379-380.

**Referee #2**

Just one suggestion, could the authors list the records of tropical cyclones in the Supporting Information? This will help the audiences to use the data and make this paper with more impacts. The authors should include more details to introduce the process of quantifying the historical records into the number.

An: We listed all data in "Supporting files" and new table for illustrations. Maybe the referee didn't find the file on system.   (cp-2018-86-supplement-version3.pdf )

---

## Author Response (AR3)

Dear Editor Chantal Camenisch,

Thank you very much for giving us very precious comments. It is really positive for this manuscript more easily to be realized for readers. Our revised corresponds to your comments in the revision version. We have changed the paragraph organization as your comments.

Dr. Huei-Fen Chen
Director and Professor,
Institute of Earth Sciences,
National Taiwan Ocean University.
e-mail: diopside0412@yahoo.com.tw